# Molecular Alterations in Sporadic and *SOD1*-ALS Immortalized Lymphocytes: Towards a Personalized Therapy

**DOI:** 10.3390/ijms22063007

**Published:** 2021-03-16

**Authors:** Isabel Lastres-Becker, Gracia Porras, Marina Arribas-Blázquez, Inés Maestro, Daniel Borrego-Hernández, Patricia Boya, Sebastián Cerdán, Alberto García-Redondo, Ana Martínez, Ángeles Martin-Requero

**Affiliations:** 1Instituto de Investigaciones Biomédicas “Alberto Sols” UAM-CSIC, Arturo Duperier 4, 28029 Madrid, Spain; marina.arribas@vet.ucm.es (M.A.-B.); scerdan@iib.uam.es (S.C.); 2Department of Biochemistry, School of Medicine, Universidad Autónoma de Madrid, Arturo Duperier 4, 28029 Madrid, Spain; 3Centro de Investigación Biomédica en Red Sobre Enfermedades Neurodegenerativas (CIBERNED), Instituto de Salud Carlos III, 28031 Madrid, Spain; ana.martinez@cisc.es; 4Institute Teófilo Hernando for Drug Discovery, Universidad Autónoma de Madrid, 28029 Madrid, Spain; 5Centro de Investigaciones Biológicas-CSIC, Avd, Ramiro Maetzu 9, 28031 Madrid, Spain; gporrasf@cib.csic.es (G.P.); ines.maestro@cib.csic.es (I.M.); pboya@cib.csic.es (P.B.); 6ALS Unit, Hospital 12 de Octubre Research Institute (i+12), 28041 Madrid, Spain; dborregohernandez.imas12@h12o.es (D.B.-H.); mito@h12o.es (A.G.-R.); 7Centro de Investigación Biomédica en Red Sobre Enfermedades Raras (CIBERER), Instituto de Salud Carlos III, 28029 Madrid, Spain

**Keywords:** amyotrophic lateral sclerosis, lymphoblast, oxidative stress, bioenergetic metabolism, NRF2, inflammation, autophagy

## Abstract

Amyotrophic lateral sclerosis (ALS) is a fatal neurological condition where motor neurons (MNs) degenerate. Most of the ALS cases are sporadic (sALS), whereas 10% are hereditarily transmitted (fALS), among which mutations are found in the gene that codes for the enzyme superoxide dismutase 1 (SOD1). A central question in ALS field is whether causative mutations display selective alterations not found in sALS patients, or they converge on shared molecular pathways. To identify specific and common mechanisms for designing appropriate therapeutic interventions, we focused on the *SOD1*-mutated (*SOD1*-ALS) versus sALS patients. Since ALS pathology involves different cell types other than MNs, we generated lymphoblastoid cell lines (LCLs) from sALS and *SOD1*-ALS patients and healthy donors and investigated whether they show changes in oxidative stress, mitochondrial dysfunction, metabolic disturbances, the antioxidant NRF2 pathway, inflammatory profile, and autophagic flux. Both oxidative phosphorylation and glycolysis appear to be upregulated in lymphoblasts from sALS and *SOD1*-ALS. Our results indicate significant differences in NRF2/ARE pathway between sALS and *SOD1*-ALS lymphoblasts. Furthermore, levels of inflammatory cytokines and autophagic flux discriminate between sALS and *SOD1*-ALS lymphoblasts. Overall, different molecular mechanisms are involved in sALS and *SOD1*-ALS patients and thus, personalized medicine should be developed for each case.

## 1. Introduction

Amyotrophic lateral sclerosis (ALS), the most common form of motor neuron disease, is a fatal neurodegenerative disease characterized by progressive muscle weakness and loss of motor neurons (MNs) in the spinal cord and brain stem. After diagnosis, the half-life is around 3 and 5 years, and the end usually occurs due to respiratory failure [1].

Compelling evidence indicates that ALS is a multisystemic disease, affecting cells other than MNs. Microglia, astrocytes, fiber muscle, and even blood cells contribute to MN degeneration [2]. On the other hand, peripheral cells from ALS patients—such as fibroblasts and lymphocytes—show traits of the disease [3,4,5]. Therefore, these patient-derived cells have been considered simple, viable, and cost-efficient cellular models of ALS.

In particular, peripheral blood cells had been previously used to study ALS pathogenesis by others and us [5,6,7,8,9]. It was shown that ALS lymphoblasts recapitulate features of affected MNs, such as alterations in TDP-43 homeostasis [5] and proven to be a suitable cell model platform to preclinical evaluation of novel TDP-43 phosphorylation inhibitors [10,11]. In this work, we have used lymphoblastoid cell lines (LCLs) derived from control and ALS patients (both sporadic and *SOD1* mutated patients). LCLs provide an unlimited material for several assays, avoiding re-sampling of vulnerable ALS patients. Nonetheless, it is believed that these LCLs resemble molecular and functional features of parent lymphocytes, based on mounting evidence from several studies, including work from our laboratory [12,13].

To date, only Riluzole and Edavarone are the approved treatments for ALS, with mild efficacy and only in some patients. Although more than 60 molecules have been investigated since Riluzole approval as a possible treatment for ALS, most of the human clinical trials failed [14]. The search for more adequate treatment options has been hampered by the highly complex etiology of the disease, with patients displaying different clinical features and progression. More than 90% of ALS cases are sporadic (sALS) [15], and about 5–10% of ALS cases are familial (fALS), with several genes and loci contributing to familial ALS [16].

In this work, we aimed to characterize at both cellular and molecular level, patient-derived LCLs, with the ultimate goal of finding specific molecular signatures enabling well-defined subgroups of ALS patients for clinical care, and as a convenient platform to test the potential efficacy of novel therapeutic compounds. To achieve this goal, we first initiated the generation of LCLs from sporadic and familial ALS patients to determine if germline mutations could define ALS susceptibility or phenotype. We then investigated well-known associated ALS hallmarks, in sALS patients and individuals carrying *SOD1* mutations. The reasons to focus on *SOD1*-mutated patients are double. First, mutations in this gene have been widely investigated [17], being found in approximately 20% of cases of fALS [7] and in approximately 5% of idiopathic/sporadic forms of the disease [18]. The second reason is due to the reduced availability of LCLs derived from carriers of mutations in *TARDBP*, *FUS*, or *C9orf72*.

Here, we focused on investigating ALS-related pathogenic mechanisms, such as mitochondrial dysfunction, oxidative stress, autophagy, and metabolic disturbances (see [17] for review). These molecular pathways could differentially contribute to the development and progression of ALS in defined subgroups of patients.

Our outcomes show that lymphoblasts from sporadic and *SOD1*-linked patients display increased rates of basal respiration and glycolytic activity but present a reduced spared respiratory capacity and glycolytic reserve, which make them vulnerable in conditions of high energy demand. On the other hand, we found important differences in the NRF2/ARE pathways when comparing cells from sporadic ALS with *SOD1*-ALS lymphoblasts. Moreover, levels of proinflammatory cytokines and autophagic flux are also different in both groups of patients.

## 2. Results

### 2.1. Oxidative Stress, Bioenergetic, and Metabolic Profiles of Lymphoblasts from Healthy Subjects, and Sporadic or SOD1 Mutation Carrier ALS Patients

Mounting evidence indicates that oxidative stress and mitochondrial dysfunction are initial events in the development and progression of neurodegenerative disorders. To address this issue, we first analyzed reactive oxygen species (ROS) and lipid peroxidation levels to monitor oxidative stress; secondly, we used the Seahorse Bioscience XF Analyzer to monitor oxygen consumption rate (OCR) and the rate of extracellular acidification (ECAR), and finally we determined changes in metabolism in lymphoblasts from healthy controls and sALS and *SOD1*-ALS mutation carriers.

The levels of ROS were measured by using the redox-sensitive fluorescent probe, 6-carboxy-2′,7′-dichlorodihydrofluorescein diacetate, di-acetoxymethyl ester (C-DCDHF-2 DA-AM). As shown in Figure 1A, basal ROS levels were significantly higher in sALS lymphoblasts compared to control cells. Thiobarbituric acid reactive substance (TBARS) assay was carried out to detect lipid oxidation [19]. Analysis of TBARS levels indicates that lymphoblast from sALS showed increased lipid peroxidation in comparison to control or *SOD1*-ALS lymphoblasts (Figure 1B).

For the Seahorse experiments, lymphoblasts obtained from four control, four sALS, and four *SOD1*-ALS individuals were pooled and incubated in the presence of 25 mM glucose, and 1 mM pyruvate as oxidizable substrates. Baseline OCR is first measured, followed by the addition of several inhibitors to estimate several parameters of mitochondrial respiration. The oligomycin sensitive OCR is calculated by subtracting the oligomycin OCR rate from baseline and indicates the fraction of OCR coupled to ATP production. The subsequent addition of the ionophore carbonyl cyanide-4-(trifluoromethoxy)phenyl hydrazone (FCCP) to uncouple the mitochondrial membrane potential increased the OCR. Maximal oxidative phosphorylation capacity can then be calculated from the FCCP-induced increase in the OCR relative to the rate reached with oligomycin treatment. The difference between maximal and baseline OCR allows determining the spare respiratory capacity. Finally, antimycin A and rotenone are added to assess non-mitochondrial respiration.

As shown in Figure 1C, both lymphoblasts from sporadic and SOD1-ALS subjects show elevated basal OCR compared with that of control cells. A small decrease in maximal respiration was observed in ALS lymphoblasts (sporadic or associated with *SOD1* mutations) relative to control cells (Figure 1C). The addition of rotenone plus antimycin C induced a reduction in OCR values to approximately 10 pmol/min/2 × 10^5^ cells in control and ALS lymphoblasts (Figure 1C). The approximately 10% difference in OCR values between oligomycin and rotenone exposure in the three groups of cells indicates no differences in oxygen consumption because of proton leakage. Figure 1D shows OCR values expressed as a percentage of basal values. It was observed a reduction of approximately 70% in OCR values in control and ALS lymphoblasts, indicating that the measured oxygen consumption was mostly driven by oxidative phosphorylation-coupled ATP synthesis. No differences were found among control and ALS cells (sporadic or *SOD1*-ALS), ruling out a direct impairment of ATP generation. The percent increase over baseline after FCCP addition was significantly lower in both lymphoblasts from sporadic and *SOD1*-ALS patients as compared with control cells, indicating a decrease in the spare respiratory capacity, which is a measure of the ATP-generating reserve to cope with an increase in ATP demand.

In addition to measuring the rate of oxygen consumption in cultured cells, Seahorse Analyzers also measure medium pH and calculate the extra acidification rate (ECAR), which is an estimation of glycolytic flux since lactic acid excretion is primarily responsible for medium acidification. As shown from Figure 1E, relative to controls, sALS and *SOD1*-ALS lymphoblasts had elevated baseline ECAR rates. As expected, the ECAR increased in control cells in response to oligomycin treatment indicating a shift to ATP production through glycolysis (Figure 1E), which is dramatically reduced in lymphoblasts derived from sporadic or *SOD1-*mutations carriers ALS patients. As illustrated in Figure 1F, the glycolytic reserve, which can be estimated by the change in the ECAR rate after oligomycin addition, is significantly lower in ALS lymphoblast. Altogether, these data show that mitochondrial function is damaged in ALS human cells.

We then carried out a metabolomic evaluation by ^1^H-NMR spectroscopy of pooled lymphoblasts from control, sALS or *SOD1*-ALS lymphoblasts to better understand possible differences in the biochemical signature of lymphoblasts derived from control or ALS (sporadic and carriers of *SOD1* mutations) individuals. To this end, we analyzed the metabolic profile of both the cell culture media and the cell pellets of the three experimental groups by ^1^H-HRMAS spectroscopy (Figure 2).

Figure 2A depicts a representative ^1^H HRMAS spectra from the extracellular medium of cultured lymphoblasts. As shown in Figure 2B, the excreted lactate, a surrogate marker of glycolysis, was found to increase in the conditioned medium by sALS or *SOD1*-ALS lymphoblasts, although values did not reach statistical significance. In contrast, we did not observe significant changes in free choline (CHO), glycerophosphocholine (GPC), or phosphocholine (PC) uptake. The intracellular metabolite profile is shown in Figure 2C. Lactate was elevated in ALS cell extracts while there were no differences in the content of CHO, GPC, or PC among the three groups of cells. In conclusion, these data are in agreement with the previously described malfunction of bioenergetic metabolism in ALS patients’ samples.

### 2.2. Different Profile of NRF2-Pathway between Sporadic and SOD1 Mutation Carriers ALS Patient’s Lymphoblast

It is proven that in response to oxidative stress, the induction of nuclear factor (erythroid-derived 2)-like2 (NRF2) stimulates expression of genes that regulate the physiological and pathophysiological outcomes of oxidant exposure. Therefore, we assessed whether the NRF2-dependent pathway was altered in ALS and if there were differences between sALS and *SOD1*-ALS patients. The analysis of NRF2 (NEF2L2) mRNA levels and NRF2-dependent genes like Heme Oxygenase 1 (*HMOX1*), NAD(P)H Quinone Dehydrogenase 1 (*NQO1*) and Thioredoxin (*TXN*) revealed that they were increased in sALS lymphoblast (Figure 3).

*NFE2L2* mRNA levels were unchanged (Figure 3A), in agreement with the concept that NRF2-pathway is mainly regulated at the protein level by disruption of its interaction with KEAP1 [20]. Thus, at the protein level, we observe that *NRF2* and *HO-1*, *NQO1*, and *TXN* are increased in the lymphoblasts of patients with sALS (Figure 4A–C). On the other hand, in lymphoblasts with mutated *SOD1* (Figure 3), we observe only a significant decrease in the levels of mRNA for *NQO1* (Figure 3C), in agreement with a previous report [21]. This opposite effect on the NRF2 pathway in *SOD1*-ALS is enhanced at the protein level, where both the levels of NRF2 and NQO1 are significantly decreased (Figure 4D,F). These data indicate important differences in the molecular mechanisms in the NRF2-pathway associated with the pathology between sALS and *SOD1*-ALS.

### 2.3. Sporadic and SOD1-Mutation Carriers ALS Patients Lymphoblast Showed Distinct Pro-Inflammatory Cytokines Profile

One of the main features present in ALS is neuroinflammation associated with infiltration of lymphocytes and macrophages [22]. It has been proposed that in ALS subtypes there are variations in survival times which correlated with cytokine levels, suggesting that ALS genetic variants have specific immune responses [23]. Therefore, to estimate the level of cytokine production by lymphocytes we determine mRNA expression levels by qRT-PCR [24]. Analysis of the mRNA expression levels of the pro-inflammatory cytokines IL-1β, IL-6, and TNF-α in the lymphoblasts of sALS and *SOD1*-ALS indicated that, while in the samples of sALS there is a significant increase of IL-1β (Figure 5A) and IL-6 (Figure 5B), in *SOD1*-ALS only an increase in TNF-α (Figure 5C) levels is observed. These results reinforce the idea that there are different pro-inflammatory cytokine profiles in the different forms of ALS.

### 2.4. Different Autophagic Profile between Sporadic and SOD1 Mutation Carriers ALS Patient’s Lymphoblasts

As one of the main stress response pathways is autophagy, we next studied autophagy in the ALS samples using lymphoblasts from 13 controls, four sALS and four *SOD1*-ALS were cultured with and without hydroxychloroquine (HCQ), an autophagy-inhibiting drug, to assess autophagy flux (Figure 6). As autophagy is a highly dynamic process, a way to study the autophagic degradation activity is by measuring the accumulation of LC3-II (autophagosome marker) in basal conditions and with a lysosomal inhibitor. The ratio between LC3-II intensities with and without HCQ reflects autophagy flux. As shown in Figure 6, sALS did not show differences with the control group, but *SOD1*-ALS samples had an increased autophagic flux. In the same way, p62, another autophagy marker was measured (Figure 6). Again, there are no differences between controls and sALS. In the case of *SOD1*-ALS patients, the difference with healthy samples was not significant, although there is a reduction of more than 26% in p62 accumulation versus the levels found in controls. p62 is an adaptor protein that binds to ubiquitinated cargo and links it to the autophagosome by its LC3-interactive region (LIR motif) [25]. As it is contained inside the autophagosome, p62 is also degraded in the lysosome. Therefore, a decrease in its protein level could be considered as related to autophagy activation [26].

To determine if this increase in autophagy in *SOD1*-ALS lymphoblasts is involved in mitochondrial clearance, TIMM23, an inner mitochondrial membrane protein was used to determine the mitochondrial mass, as an indirect measure of mitophagy (Figure 6). Surprisingly, although there are no differences between controls and *SOD1*-ALS, TIMM23 is significantly increased in these lymphoblasts compared to sALS. Therefore, these results underline the idea of an increase in bulk autophagy in *SOD1*-ALS.

## 3. Discussion

To date, there is no effective pharmacological treatment for ALS. The pathobiology of ALS has proven complex and multisystemic. Therefore, the understanding of the molecular and cellular mechanisms driving neurodegeneration is imperative for developing innovative and effective therapies, which could delay disease onset, slow its progression and/or prolong a patient’s survival in a personalized manner considering different ALS etiologies. This work presents, for the first time, the description of alterations on common molecular mechanisms between sALS and *SOD1*-ALS, as well as specifically deregulated mechanisms in each one.

Here, we investigated several molecular disease mechanisms in sALS and evaluated whether mutations in the *SOD1* gene affect differentially ALS-related pathways. Our outcomes show enhanced ROS levels, and increased lipid peroxidation mainly on sporadic ALS lymphoblasts following the idea that oxidative stress plays a critical role in the pathogenesis of neurodegenerative disorders, including ALS [27]. Lipid peroxidation injures phospholipids and oxidized phospholipids can play an important role in many inflammatory diseases and frequently mediate proinflammatory change [28,29]. These results are in agreement with our findings that only in sALS lymphoblasts we observed an increase in the antioxidant NRF2 pathway and the inflammatory cytokines IL1-β and IL-6, as a possible response to lipid peroxidation.

Patient-derived lymphoblasts had higher mitochondrial respiration than control cells under basal conditions, although the ATP coupling efficiency, that is the percentage of OCR due to ATP synthesis, was found to be similar in the three experimental groups. In agreement with a previous report [9], we observed a marked reduction of the spare respiratory capacity, in both sALS and *SOD1*-ALS lymphoblasts, which suggest the risk for ALS patient-derived cells to cope with situations demanding high energy production. Despite the increase in OCR-linked ATP synthesis, ALS lymphoblasts show elevated rates of extracellular acidification and lactate production than control cells. Elevated mitochondrial respiration in tandem with increased glycolytic capacity had been reported in fibroblasts from ALS patients [30], suggesting general upregulation of metabolism. The higher levels of lactate production by ALS cells are consistent with the increase in aerobic glycolysis (Warburg effect) that has been noted in ALS [31]. Of note is the fact that ALS cells (both sporadic and *SOD1* mutated) had a reduced glycolytic reserve, compared to control cells, which may also compromise the response to high energetic demand for ATP production under stressful conditions. These results can be also explained assuming mitochondrial damage in ALS cells. In that case, an increase of ROS would be produced by damaged mitochondria while the glycolytic pathway would be used to obtain ATP.

One of the main mechanisms that the cell has to defend itself against oxidative stress is the activation of the antioxidant system NRF2/ARE [20]. In recent years, it has been discovered that NRF2 is a pleiotropic transcription factor, becoming a crucial regulator of the cellular defense against oxidative stress and xenobiotics, but also involved in processes such as autophagy, among others [32]. Concerning neurodegenerative diseases, it has been observed that the NRF2 pathway can be compromised [33,34,35,36,37]. Regarding ALS, it has been described that NRF2 mRNA and protein levels were reduced in ALS patients relative to control tissues, [38,39], but NRF2-dependent genes were not evaluated. Moreover, an additional study indicates that there is a lack of association between NRF2 promoter gene polymorphisms and oxidative stress biomarkers in ALS patients [40]. To get further insight into this pathway, we assessed the status of the NRF2/ARE system in sALS and *SOD1*-ALS lymphoblast cells. We observed a different pattern of the NFR2 pathway (Figure 3 and Figure 4). Our results indicate that NRF2 levels are not regulated at the transcriptional level either in sALS or *SOD1*-ALS cells. In sALS cells, NRF2 is altered at the protein level, a finding consistent with one of the main NRF2 modulation pathways, the interaction with the negative regulator Kelch-like ECH-associated protein 1 (KEAP1) [20,32]. In sALS lymphoblasts, there is a significant increase in NRF2-dependent genes like HMOX1, NQO1, and TXN, at mRNA and protein levels. Heme oxygenase-1 (HO-1) is an enzyme that catalyzes the degradation of heme, which produces biliverdin, ferrous iron, and carbon monoxide with antioxidant, anti-inflammatory, anti-apoptotic, anti-proliferative, and immunomodulatory effects [41]. Moreover, NQO1 and TXN are key antioxidant systems through modulation of substrates including ubiquinone, vitamin E quinone, and superoxide [42] and regulating protein dithiol/disulfide balance [43], respectively. Our results indicate that in sALS, the increase in ROS levels induces the antioxidant response mediated by the NRF2 pathway, but it is not enough to restore ROS to control levels, despite the activation of all these antioxidant enzymes. On the other hand, in *SOD1*-ALS lymphoblast, practically no changes are observed in mRNA levels, except for *NQO1*, where there is a very significant reduction (Figure 3). At the protein level, there is a decrease in both the levels of NRF2 and NQO1. These results clearly show a significant difference in the molecular mechanisms involved in the pathology of the disease concerning the antioxidant system NRF2, depending on the type of ALS. Therefore, the pharmacological modulation of NRF2 as a therapeutic strategy for ALS should be personalized, based on the molecular alterations displayed by the different type of patients.

Autophagy is a general stress response and plays a general cytoprotective role in several pathogenic conditions including neurodegenerative diseases. Autophagy activity in ALS has been attracting more attention due to its role in the degradation of protein inclusions or defective mitochondria. As these organelles have been reported to be accumulated in ALS models, autophagy and mitophagy are thought to be blocked [44]. Moreover, mutations in autophagy or mitophagy genes have been related to ALS [45]. We thus performed the analysis of the autophagy-related proteins LC3 and p62 (Figure 6). Autophagy flux, measured by the blockage with the lysosomal inhibitor HCQ, showed an increase in *SOD1*-ALS patients compared to control. In the case of sALS, no differences were reported compared to control, and more variability between samples was found, with one of the patients showing a huge increase in autophagy flux, which could be used in a more personalized medicine approach. To further confirm the increase in autophagy flux in *SOD1*-ALS, the protein level of p62 was also measured. As it is shown, there are no significative differences, but a tendency to be lower in *SOD1*-ALS compared to control. Moreover, p62 has been related to NRF2 due to its interaction with KEAP1. When p62 is accumulated due to autophagy blockage, it interacts with KEAP1, interfering with the degradation directed by KEAP1 of NRF2 [46]. Therefore, the decrease in NRF2 protein level in *SOD1*-ALS shown in Figure 4, could agree with the tendency of p62 to be lower in those samples compared to controls (Figure 6).

Then, to determine if this increase in bulk autophagy is mainly mitophagy, we also evaluated the levels of the inner mitochondrial membrane protein TIMM23. No differences were found between *SOD1*-ALS and controls, but a significative accumulation of mitochondria was reported in the *SOD1*-ALS samples in comparison with sALS. Thus, it is difficult to conclude at present a mitophagy blockage in *SOD1*-ALS samples, but apparently, the increase in autophagy flux in these samples is not related to the selective degradation of mitochondria. Interestingly, a recent work done in peripheral blood mononuclear cells (PBMCs) from ALS patients, this mitophagy blockage was present in combination of an increase in bulk autophagy [47]. It is tempting to speculate that the selective degradation of mitochondria and protein aggregates be blocked while bulk autophagy is increased to counteract the defects in the selective processes. Our data are also in agreement with the increase in autophagy markers reported in skin fibroblasts from patients with mitochondrial diseases but without an increase in mitophagy [48]. These results showed an apparent increased in autophagy in *SOD1*-ALS patients, without changes in mitochondrial content, as a compensatory process to avoid oxidative stress. Regarding sALS, no difference in autophagy profile has been reported compared to control samples. Nevertheless, autophagy is a challenging process to be studied in further analysis, to confirm an increase in bulk autophagy in *SOD1*-ALS, to characterize this process further to be able to target it as a potential therapeutic approach in the treatment of ALS.

Altogether, these results can throw some light on the future of personalized treatments for ALS and in clinical trial stratification. Antioxidants would be a general treatment for sALS patients and, despite the controversial raised with their use in the past, the last drug approved by some agencies for ALS treatment, Edaravone, is a repurposed antioxidant agent [49]. Furthermore, our results have shown different pro-inflammatory cytokine profiles for sporadic and *SOD1-*ALS subtype which agrees with recent data that also shown a negative association with the survival rate [23]. Besides, the use of non-aspirin-NSAIDs and acetaminophen, which decrease the risk for the development of ALS [50], would be especially useful for sALS where high levels of cytokines have been found. On the other hand, decreased levels of NRF2 in *SOD1*-ALS points to dimethyl fumarate (DMF) [51,52], an inducer of this pathway, as a good therapy for this sub-group of patients. As probably the efficacy of this drug will be higher in *SOD1*-ALS patients, stratification data analysis of the ongoing TEALS phase II clinical trial [53] where ALS patients are being treated with DFM, would be recommended. Finally, and regarding autophagy, compounds—such as rapamycin—that increase autophagic flux would be more efficacious in restoring the homeostasis of this key cellular function in sALS than in the *SOD1* subtype. This observation would be useful for patient stratification analysis in the ongoing RAP-ALS clinical trial [54].

In summary, our results show common and different disease mechanism between sALS and *SOD1*-ALS. The group-specific changes described here, could serve to decide more specific and personalized pharmacological treatments. However, further work is needed with bigger cohorts of patients to validate these different molecular features as truly subtype-disease markers. Nonetheless, it is envisioned that assessment of NRF2 activity, cytokine levels, or autophagy flux in peripheral cells—easily accessible from patients—could eventually serve to track the efficacy of particular therapeutic strategies in each group of patients. Additional work to demonstrate the usefulness of patient-derived LCLs to follow the patient response to a particular treatment, as well as to extensively characterize LCLs from patients carrying mutations in *TARDBP, Corf72*, among other ALS-related genes is underway in our laboratory.

## 4. Materials and Methods

### 4.1. Materials

RPMI 1640 culture medium Cat#: Lo500 was obtained from Biowest/Labclinics (Barcelona, Spain), Fetal Bovine Serum (FBS) cat no. F7524 was purchased in (Merck, Madrid, Spain).

### 4.2. Lymphoblastic Cell Lines

A total of 24 subjects participated in this study, 6 sporadic ALS patients (age range 55–76, 3 males and 3 females), four male *SOD1*-ALS patients, (age range 46–54) and 14 healthy subjects (age range 52–88, 7 females, 7 males). The detailed demographic and relevant clinical data of all subjects are shown in Table 1.

Participants or their relatives gave written informed consent. This study was approved by the Hospital Doce de Octubre and the Spanish Council of Higher Research Institutional Review Boards. All patients were diagnosed by applying the revised El Escorial criteria [55]. Genetic testing for *SOD1*, *TARDBP*, *FUS*, and *C9orf72* was performed in all the sporadic cases. Cases E18.5 and E18.6 were analyzed with a panel including 48 ALS-related genes. The full exome of cases E11, and E18.7 were sequenced. Control healthy individuals were recruited separately and did not have any known neurological disorder.

Blood samples from most of the participants were taken at diagnosis and therefore were not receiving any treatment. Cases E18.5 (ALSRF-r 48), E18.6 (ALSRF-r 39), and E18.7 (ALSRF-r 41) were already treated with Riluzole, when blood samples were taken.

The lymphoblastoid cell lines (LCLs) were generated by infecting peripheral blood lymphocytes with the Epstein–Barr virus as previously described [56]. LCLs were grown in suspension in T flasks, in RPMI-1640 medium containing 2 mM L-glutamine, 100 μg·mL^−1^ streptomycin/penicillin and 10% (*v*/*v*) fetal bovine serum (FBS) and maintained in a humidified 5% CO_2_ incubator at 37 °C.

### 4.3. Measurement of Reactive Oxygen Species (ROS) and Thiobarbituric Acid Reactive Substances (TBARs)

The intracellular accumulation of ROS was determined using the fluorescent probe CM-H2DCFDA (Invitrogen, C6827). Control and ALS lymphoblasts were loaded with 10 µM CM-H2DCFDA for 30 min. Fluorescence measurements were carried out using a POLARstar Galaxy spectrofluorimeter (BMG Labtechnologies, Offenburg, Germany) at λex/em λ495/510 nm. Lipid peroxidation was determined as the formation of thiobarbituric acid-reactive substances (TBARS), according to a previous report [57]. 6 × 10^6^ cells were centrifugated and pellets were resuspended in 200 µL PBS and 400 μL of TBA reagent (0.375 g TBA, 7.5 g trichloroacetic acid, and 2.54 mL HCl) were added and incubated at 95 °C for 30 min. A pink chromophore was produced in samples in direct relation to the amount of peroxidized products. Samples were then kept in ice for 5 min and centrifuged at 3000× *g* for 15 min. The optical density of the supernatants was measured in a spectrometer at 532 nm. The amount of TBARS (mostly malondialdehyde [MDA]) was calculated by interpolation of values in a constructed MDA standard curve with 1,1,3,3-tetrametoxypropane, and results were expressed as nanomoles of MDA per cells.

### 4.4. Oxygen Consumption and Extracellular Acidification Rates Measurements

We used the Extracellular Flux Analyzer XFp (Agilent Seahorse) to evaluate energy metabolism. 2 × 10^6^ cells/mL, (pools of four different patients’ LCLs in each group) were cultivated in Seahorse XFp miniplates coated with 3.5 µg Cell-Tak/well (Fisher Scientific) and supplemented with alpha-MEM Stock A. at 37 °C, 5% CO_2_, 95% air for 16 h to allow adherence of cells to the surface of the wells. Assessment of the bioenergetic profiles of control and ALS lymphoblasts was performed using the Mito Stress Test Kit (Agilent Technologies for Seahorse XFp). On the day of the analysis, the culture medium was changed to 180 μL of bicarbonate-free DMEM (Sigma-Aldrich) supplemented with glucose 5 mM, 2 mM L-glutamine, 1 mM pyruvate, 2% FBS (Sigma-Aldrich), and HEPES 5 mM pH 7.4. Cells were maintained for 1 h in a CO_2_-free incubator. The mitochondrial respiration and glycolytic activity were determined from the oxygen consumption rate (OCR) and extracellular acidification rate (ECAR), respectively.

The experimental setup was based on a predesigned Agilent’s program with slight modifications. Once four measurements under basal conditions were taken, oligomycin 1 μM was added to inhibit mitochondrial ATP synthesis. Subsequently, measurements were made in the presence of the uncoupler FCCP (carbonylcyanide-p-trifluoromethoxy-phenylhydrazone, 3 μM) to determine the maximum respiratory rate. Finally, three measurements were made in the presence of rotenone and antimycin (2.5 μM of each inhibitor) to inhibit mitochondrial respiration.

### 4.5. HRMAS-NMR Acquisition and Processing

Before High-Resolution Magic Angle Spinning (HRMAS) nuclear magnetic resonance (NMR) analysis, each cell pellet was flushed with D_2_O to remove the residual water and to improve suppression of the water resonance [58]. The sample was introduced in a HRMAS zirconium rotor (4 mm OD) fitted with a 50 µL cylindrical insert, and transferred into the Magic Angle Spinning (MAS) probe, cooled to 5 °C. HRMAS spectra were acquired on a 11.7 Tesla MHz Bruker AVANCE WB Spectrometer operating at 500.13 MHz for ^1^H, at 5 °C and 4 kHz spinning rate. Two types of 1D ^1^H HRMAS spectra were acquired on each sample: (i) a water presaturated spectrum (5 s relaxation delay) using a pulse-and-acquire sequence (zgcppr) with π/2 pulses, 10 kHz spectral width, 32k data points, and 128 scans; (ii) a Carr-Purcell-Meiboom-Gill (CPMG) spectrum using a spin-echo sequence (cpmgpr) with water presaturation (5 s) during relaxation delay, 1 ms echo time, and 144 ms total echo time (2nτ), 32k data points and 128 scans. ^1^H HRMAS resonances areas were quantified on CPMG spectra using MestRec Nova software (https://mestrelab.com/software/mnova/nmr accessed on 5 February 2020) and expressed as percent of the total area of the ^1^H HRMAS spectrum.

### 4.6. Analysis of mRNA Levels by Quantitative Real-Time PCR

Total RNA extraction, reverse transcription, and quantitative polymerase chain reaction (PCR) was done as detailed in the previous article [59]. Primer sequences are shown in Appendix A. Data analysis was based on the ΔΔCT method with normalization of the raw data to housekeeping genes (Applied Biosystems). All PCRs were performed in triplicates.

### 4.7. Immunoblotting

Whole-cell lysates were prepared in RIPA-Buffer (25 mM Tris-HCl pH 7.6, 150 mm NaCl, 1 mm EGTA, 1% Igepal, 1% sodium deoxycholate, 0.1 % SDS, 1 mm PMSF, 1 mm Na_3_VO_4_, 1 mm NaF, 1 μg/mL aprotinin, 1 μg/mL leupeptin, and 1 μg/mL pepstatin). Whole-cell lysates containing 25 μg of whole proteins from lymphoblast cells were loaded for SDS-PAGE electrophoresis. Immunoblots were performed as described in [60]. The primary antibodies used are described in Appendix A. To study autophagy flux, cells were treated with 30 µm hydroxychloroquine (HCQ) for the last 3 h of incubation before lysing. Proteins were extracted with 200 μL lysis Buffer (50 mM Tris-HCl pH 6.8, 10% glycerol (*v*/*v*) and 2% sodium dodecyl sulfate (SDS) (*w*/*v*) with protease inhibitors 1x (Sigma, P8783), phosphatase inhibitors (1 mm sodium orthovanadate (Sigma, S6508), 1 mm sodium fluoride (Sigma, 201154), and 5 mm sodium pyrophosphate decahydrate (Sigma, 221368)). 20 μg of protein were loaded in CriterionTM TGX Precast Midi Protein gels (BioRad, 5671124) and transferred to PVDF membranes (BioRad, 170–4157). The primary antibodies used are described in Appendix A.

### 4.8. Statistical Analyses

Data are presented as mean ± SEM. To determine the statistical test to be used, we employed GraphPad Instat 3, which includes the analysis of the data to normal distribution via the Kolmogorov–Smirnov test. Also, statistical assessments of differences between groups were analysed (GraphPad Prism 5, San Diego, CA, USA) by unpaired Student’s *t*-tests when normal distribution and equal variances were fulfilled, or by the non-parametric Mann–Whitney *U*-test. One and two-way ANOVA with post hoc Newman–Keuls test or Bonferroni’s test were used, as appropriate.

## 5. Conclusions

Our results support the usefulness of ALS patients-derived lymphoblastoid cell lines to investigate disease mechanisms, showing features of degenerating MNs such as mitochondrial dysfunction, or oxidative stress. More importantly, our data unveiled ALS subtype-specific alterations. In particular, NRF2 activity and autophagic flux appear to be differentially regulated in sporadic or *SOD1*-mutant cells, respectively. Thus, assessment of the molecular changes described herein, in easily accessible peripheral cells from patients, can help for the correct stratification of patients in the design and analysis of clinical trials, as well as the search for more effective and tailored therapeutic strategies.

## Figures and Tables

**Figure 1 ijms-22-03007-f001:**
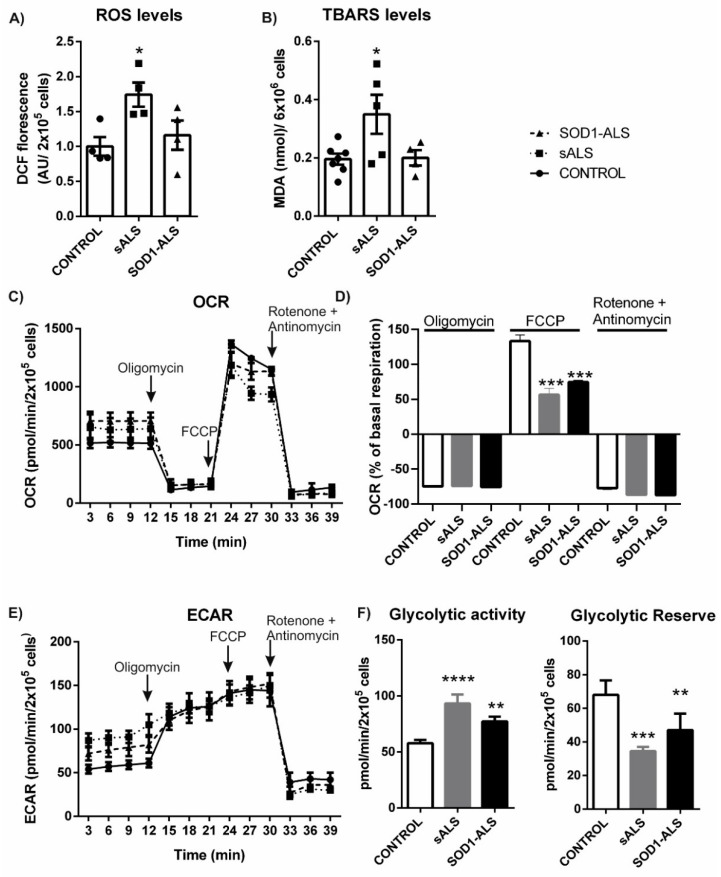
Reactive Oxigen Species (ROS) levels and bioenergetic profile of lymphoblasts derived from control individuals, sporadic ALS and *SOD1-*ALS. (**A**) Four lymphoblastoid cell lines from the three experimental groups were seeded at a concentration of 1 × 10^6^ xml^−1^ in RPMI medium containing 10% FBS for 24 h. Subsequently, about 200,000 cells were incubated with the CM-H_2_DCFDA probe for 30 min. The data show the mean ± the standard error of the mean of four LCLs in each group in three independent experiments. * *p* < 0.05. (**B**) Thiobarbituric acid-reactive substances (TBARs) representing malondialdehyde (MDA), resulting from oxidation of lipid substrates. Dots indicate the mean of *n* = 7 (controls), *n* = 5 (sALS), and *n* = 4 (*SOD1*-ALS) samples ± SEM. Asterisk denotes significant differences * *p* < 0.05, comparing the indicated groups with the basal condition according to a one-way ANOVA followed by Tukey post-test. (**C**) For these experiments, a pool was made with lymphoblasts derived from four control individuals and two more with lymphoblasts from four patients with sporadic ALS or *SOD1*-ALS. 200,000 cells were seeded per well on the Seahorse XF24 Analyzer. The cells were cultured for 24 h, before measurements were made in an unbuffered DMEM medium at pH 7.4 containing pyruvate at 1 µM and glucose at 25 µM. The graph represents the respiration of control cells and ALS cells over time and the effect of the addition of 1 µM oligomycin, 3 µM FCCP and 1 µM rotenone + 1 µM antimycin. The data show the mean of four experiments ± the standard error of the mean. (**D**) Percent change in mitochondrial respiration in response to mitochondrial inhibitors. Bars represent the mean change in OCR from baseline ± SEM (*** *p* < 0.001 as compared with control). (**E**) Temporal assessment of extracellular acidification rate (ECAR). (**F**) Quantitative measurement of glycolytic capacity and glycolytic reserve (change in ECAR after oligomycin injection) respectively. ***p* < 0.01; *** *p* < 0.001; **** *p* < 0.0001.

**Figure 2 ijms-22-03007-f002:**
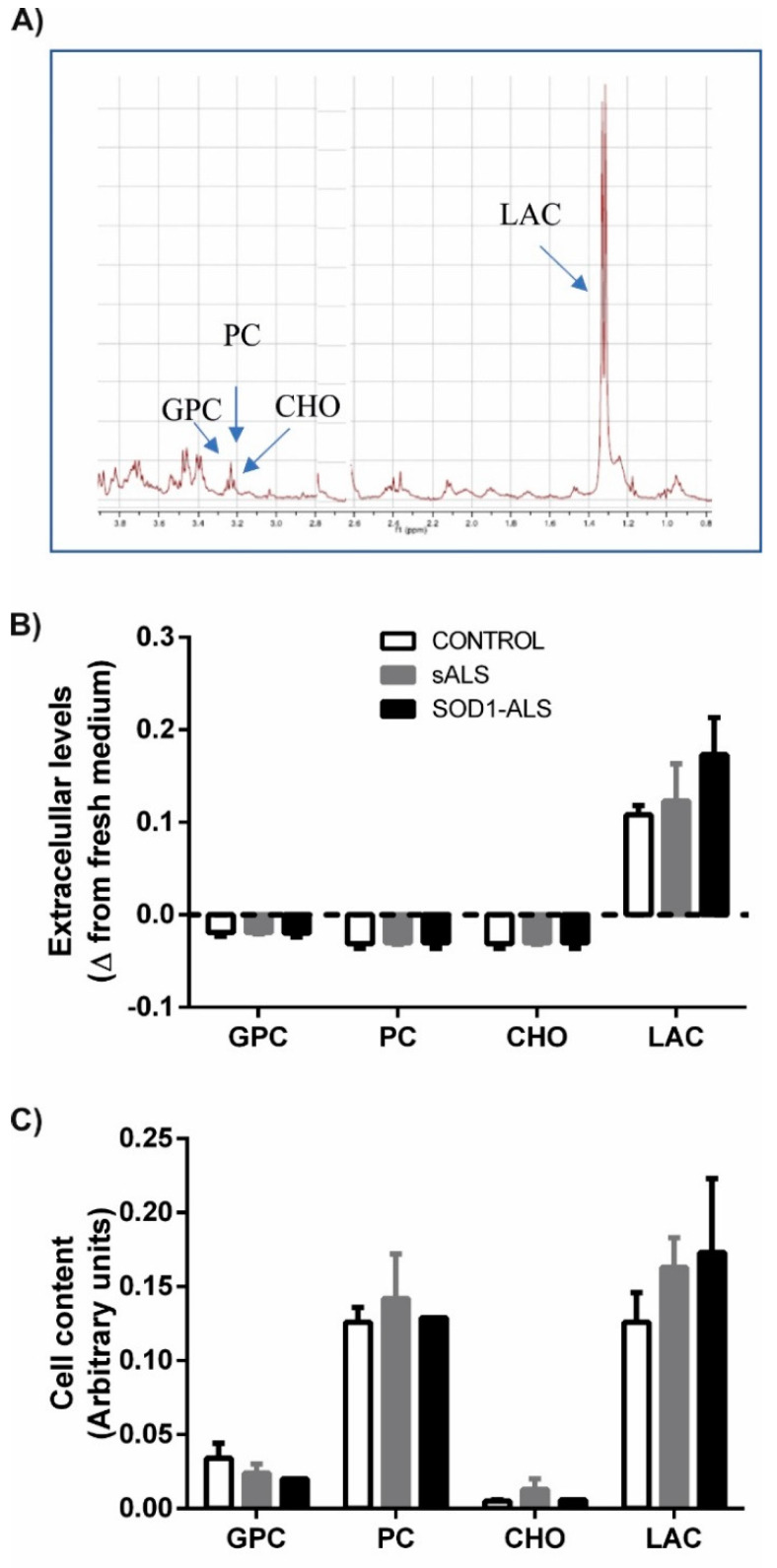
Metabolic profiles of lymphoblasts from healthy subjects, and sporadic or *SOD1* mutation carriers ALS patients. (**A**) Representative spectrum of 1H nuclear magnetic resonance. The peaks of the metabolites of interest are marked with arrows: GPC, PC, CHO and LAC. Abbreviations: GPC, glycerol phosphocholine; PC, phosphatidylcholine; CHO, choline; DMSO, dimethyl sulfoxide; LAC, lactate. (**B**,**C**) Lymphoblasts (pool of 4 individuals) of controls, patients with sALS and *SOD1*-ALS were incubated for 24 h in RPMI medium containing 10% FBS. The cell pellets and extracellular medium were collected after centrifuging at 10,000 rpm for 10 min and metabolites were determined by 1H NMR spectroscopy. The changes in the concentrations of the different metabolites in cell medium (**A**) or changes in intracellular content (**C**) medium are shown. All values are means ± SEM of 4 experiments, except when SEM is not indicated because only two measurements were available.

**Figure 3 ijms-22-03007-f003:**
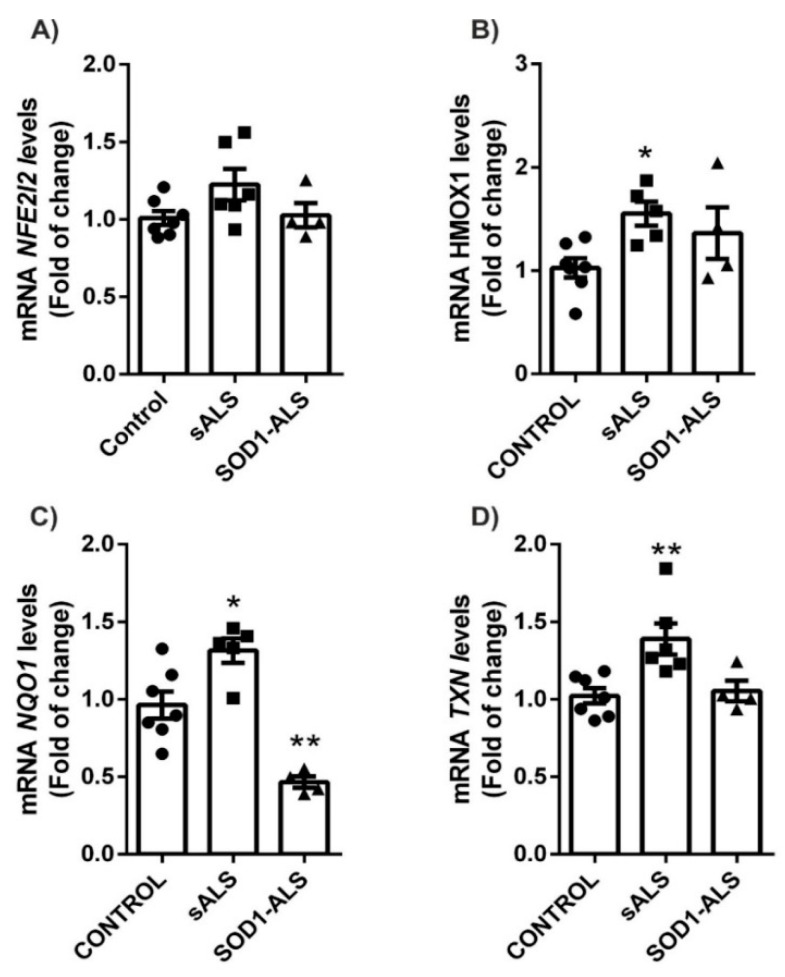
sALS lymphoblasts showed increased mRNA expression levels of NRF2-pathway. Quantitative real-time PCR determination of messenger RNA levels of *NFE2L2* (**A**) and NRF2-regulated genes coding *HMOX1* (**B**)*, NQO1* (**C**), and *TXN* (**D**), normalized by *β-ACTIN* messenger RNA levels. Dots indicate the mean of *n* = 6–7 (controls), *n* = 5–6 (sALS) and *n* = 4 (*SOD1*-ALS) samples ± SEM. Asterisks denote significant differences * *p* < 0.05 and ** *p* < 0.01, comparing the indicated groups with the basal condition according to a one-way ANOVA followed by Tukey post-test.

**Figure 4 ijms-22-03007-f004:**
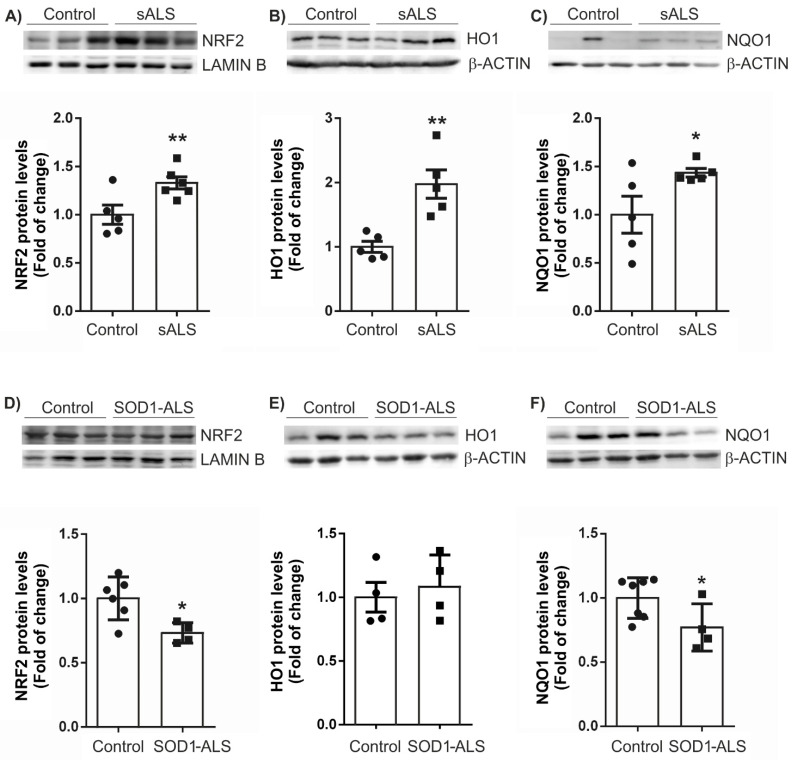
sALS and *SOD1*-mutant ALS lymphoblasts revealed significant differences in the NRF2 signaling pathway at the protein level. Immunoblot analysis in whole control and sALS cell lysates of protein levels of NRF2 (**A**) and LAMIN B as a loading control, HO-1 (**B**), NQO1 (**C**), and β-ACTIN as a loading control. Densitometric quantification of representative blots normalized for β-ACTIN. Dots indicate the mean of *n* = 5 samples ± SEM. Immunoblot analysis in whole control and *SOD1*-ALS cell lysates of protein levels of NRF2 (**D**), HO-1 (**E**), NQO1 (**F**), and β-ACTIN as a loading control. Densitometric quantification of representative blots normalized for β-ACTIN. Dots indicate the mean of *n* = 5-6 (control) and *n* = 4 (*SOD1*-ALS) samples ± SEM. Asterisks denote significant differences * *p* < 0.05 and ** *p* < 0.01, comparing the indicated groups with the basal condition according to the Student’s *t*-test.

**Figure 5 ijms-22-03007-f005:**
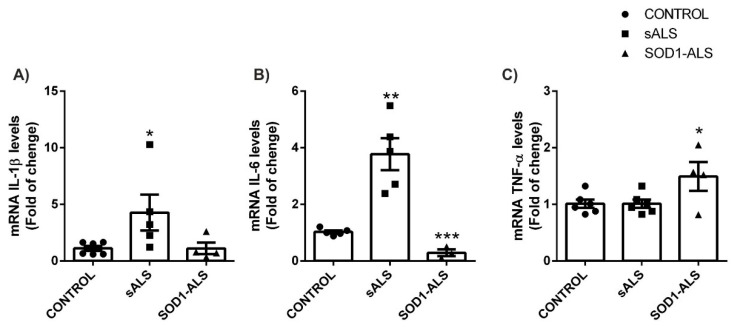
Increased IL-1β and IL-6 levels mRNA expression levels in sALS and *SOD1*-ALS lymphoblasts. Quantitative real-time PCR determination of messenger RNA levels of pro-inflammatory cytokines IL-1β (**A**), IL-6 (**B**), and TNF (**C**), normalized by β-ACTIN messenger RNA levels. Dots indicate the mean of *n* = 6 (controls), *n* = 5–6 (sALS) and *n* = 4 (*SOD1*-ALS) samples ± SEM. Asterisks denote significant differences * *p* < 0.05 and ** *p* < 0.01, comparing the indicated groups with the basal condition according to a one-way ANOVA followed by Tukey post-test.

**Figure 6 ijms-22-03007-f006:**
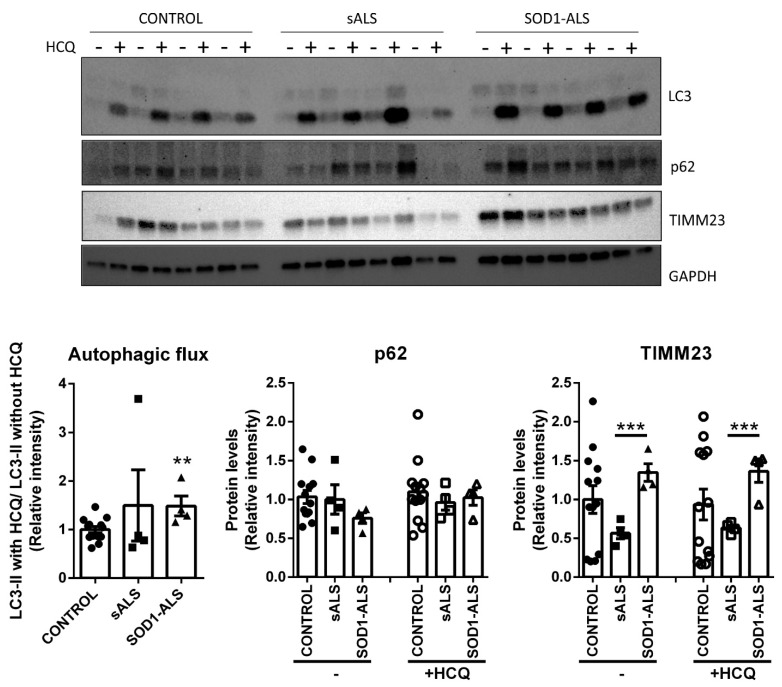
Increased autophagy flux in *SOD1*-ALS lymphoblasts. Densitometric quantification of representative blots normalized for GAPDH. Immunoblot analysis in whole control, sALS and *SOD1*-ALS cell lysates of protein levels of LC3, p62, and TIMM23. Dots indicate the mean of *n* = 13 (control) *n* = 4 (sALS) and *n* = 4 (*SOD1*-ALS) samples ± SEM. Asterisks denote significant differences ** *p* < 0.01 and *** *p* < 0.001.

**Table 1 ijms-22-03007-t001:** Demographic and clinical characteristics of the subjects. * Negative for *SOD-1*, *TARDBP*, and *C9ORF72.*

Patient ID	Clinic Presentation	Affected Motoneurons	Age	Gender	Mutation
C-34	n/a		88	Female	----
C-35	n/a		63	Male	----
C-37	n/a		57	Female	----
C39	n/a		53	Female	----
C105	n/a		54	Female	----
C106	n/a		67	Female	----
C107	n/a		58	Female	----
C108	n/a		68	Female	----
C110	n/a		75	Male	----
C112	n/a		71	Male	----
C122	n/a		77	Male	----
C123	n/a		55	Male	----
C132	n/a		52	Male	----
E2	Bulbar	MNS	76	Female	*
E4	bulbar	MNS & MNI	54	Female	*
E5	Spinal	MNS	54	Female	*
E6	Bulbar	MNI	79	Male	*
E8	Respiratory	MNI	55	Male	*
E10’	Bulbar	MNS	68	Male	*
E11	Spinal	MNI	64	Male	*SOD1* het N65S
E18-5	Spinal	MNI	46	Male	*SOD1* het p.Leu117Val
E18-6	Spinal	MNS & MNI	58	Male	*SOD1* het p.ASn139His
E18-7	Spinal	MNI	59	Male	*SOD1* het p.Leu117Val

## Data Availability

The datasets analyzed during the present study are available from the corresponding author on reasonable request.

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
