# Peer review of "Molecular Alterations in Sporadic and *SOD1*-ALS Immortalized Lymphocytes: Towards a Personalized Therapy"

_ijms, 2021, doi:10.3390/ijms22063007_

Round 1

Reviewer 1 Report

The authors investigated some pathways usually involved in ALS (i.e. mitochondrial dysfunction, Nrf2 pathway, cytokines profile, and autophagic flux) on a patient-derived cell model (LCLs). 

I have some concerns:

  1. The experimental plan and the aim of the work is not clear. What the authors want to demonstrate? If they want to demonstrate that LCLs are a good model to study ALS, they should compare LCLs with other models. Moreover, I did not understand why the authors prefer to investigate some pathways respect to other pathways.
  2. The experimental plan is not always correct. For example the increase of LC3-II is not always related to the increase of the autophagic flux. Its increase sometimes suggests the inhibition of the autophagic flux. Thus, further investigation must be carried out.
  3. The data obtained are not always supported by the literature. For example the basal OCR level is extremely high compared to other article on the same cell model. I think that the experiment was not carried out in the correct way. Another example is the qPCR to investigate cytokines level. I would suggest an ELISA to investigate the real level of cytokines in the medium of LCLs.
  4. The article lacks of originality and novelty.

I suggest the authors to focus on one of the proposed pathway in order to improve the significance of their work.

Author Response

Response to Reviewer #1

  1. The experimental plan and the aim of the work is not clear. What the authors want to demonstrate? If they want to demonstrate that LCLs are a good model to study ALS, they should compare LCLs with other models. Moreover, I did not understand why the authors prefer to investigate some pathways respect to other pathways.

We have tried to clarify the main objective of our work in the revised version of our Ms.

The ultimate goal of our work is to open avenues for disease-modifying therapies in ALS patients in a personalized manner based on potential differences in molecular disease mechanisms that might exist between sporadic or familial cases. To address this issue, we have focus on investigating some of the well-known molecular pathways involved in the ALS etiology (i.e mitochondrial dysfunction, oxidative stress, autophagic flux etc.). We have used lymphoblastoid cell lines derived from patients, as it is recognized the usefulness of this cell model for functional studies even in neurodegenerative diseases.

In fact, we have recently described that lymphoblasts from FTLD-TDP and sporadic ALS patients recapitulate pathogenic features of TDP-43 homeostasis (increased phosphorylation, truncation and altered subcellular location) observed in affected neurons (Posa et al. Molecular neurobiology 2019, 56, 2424-2432; Vaca et al., Journal of neurochemistry, 10.1111/jnc.15118, 2020).

  1. The experimental plan is not always correct. For example the increase of LC3-II is not always related to the increase of the autophagic flux. Its increase sometimes suggests the inhibition of the autophagic flux. Thus, further investigation must be carried out.

We thank the referee for this remark. In this experiment we have carried out the LC3 turnover assay to asses autophagy flux using the lyososomal inhibitor chloroquine. Accordingly, the differences in the amount of LC3-II between samples in the presence and absence of lysosomal inhibitors represent the amount of LC3 that is delivered to lysosomes for degradation (i.e., autophagic flux) see; Methods in mammalian autophagy research. Cell. 2010 Feb 5;140(3):313-26. PMID: 20144757 and Guidelines for the use and interpretation of assays for monitoring autophagy (3rd edition). Autophagy. 2016;12(1):1-222. PMID: 26799652.

  1. The data obtained are not always supported by the literature. For example the basal OCR level is extremely high compared to other article on the same cell model. I think that the experiment was not carried out in the correct way. Another example is the qPCR to investigate cytokines level. I would suggest an ELISA to investigate the real level of cytokines in the medium of LCLs.

The concern of Reviewer #1 regarding the higher rates of basal OCR described here compared with previous reports might not be justified, as we expressed our results as pmol/min/2 x10-5 cells, while for example in Pansarasa paper (Disease Model and Mechanisms 2018 11), basal OCR rates were expressed in pmol/min without indicating the number of cells or protein content in each well.

Regarding qPCR to investigate cytokines level, we greatly appreciate this comment. It has been described that depending on the specific aim of an investigation, different methods could be chosen (Favre N, Bordmann G, Rudin W. Comparison of cytokine measurements using ELISA, ELISPOT and semi-quantitative RT-PCR. J Immunol Methods. 1997 May 12;204(1):57-66. doi: 10.1016/s0022-1759(97)00033-1. PMID: 9202710.). Amsen et al (Amsen D, de Visser KE, Town T. Approaches to determine expression of inflammatory cytokines. Methods Mol Biol. 2009;511:107-142. doi:10.1007/978-1-59745-447-6_5) indicated that “real-time quantitative polymerase chain reaction (Q-PCR), involves measurement of cytokine mRNA transcript abundance. This method is relatively straightforward and quantitative and allows for the detection of many different cytokines from relatively small sample amounts”.

Lymphoblast cultures are in suspension, in a high amount of medium. To obtain the cells, the culture is centrifuged and the pellet is used, from which the RNA is extracted. Our main concern when measuring cytokine levels in the medium is that actual cytokine levels may be underestimated. Therefore, we used qPCR in this study. A short description has been inserted in Results section.

  1. The article lacks of originality and novelty

This work presents for the first time the description of specifically deregulated disease mechanisms between sALS and SOD1-ALS. NRF2 pathway, autophagic flux and levels of cytokines clearly discriminate sporadic and SOD1-associated ALS patients. Therefore, our results can help to stratification of patients for distinct therapeutic strategies in each case. We believe that it is an important aspect of the research and that it broadens the perspective of lymphocyte application for monitoring the efficacy of candidate drugs.

Reviewer 2 Report

In this study, the authors have compared the respiratory and glycolytic activity, the ROS generation, NRF2 pathway and autophagy markers in lymphoblasts from patients with sporadic ALS and familial ALS with SOD1 mutations. From the data they conclude that lymphoblasts from both sALS and SOD1-ALS patients exhibit increased rates of basal respiration and glycolytic activity. However, they conclude from the mRNA and immunoblot data that there are differences between the sALS and SOD1-ALS in the levels of antioxidant NRF2, inflammatory cytokines and autophagy markers. They claim that the molecular differences in the lymphoblasts of sALS and SOD1-ALS are sufficiently significant to justify distinct treatment for these two groups especially for drugs targeting oxidative stress and autophagy homeostasis.      

There are several concerns with this study:

  1. Considering the possibility to work with iPSC-derived motor neurons from ALS patients, lymphoblasts are not the ideal cell type to compare defective molecular mechanisms causative of ALS disease.
  2. The number of ALS patients in the sporadic (6) and familial SOD1 (4) is very low. Considering the large variations in some of the results including in the controls, the significance of the findings is questionable.
  3. The data are generally of low quality. In Fig. 4B, the immunoblot suggests great variations in levels of HO1 which is not reflected in the quantification of the graph.  I see great variation in the band intensities in controls and sALS. In the controls the variation is not reflected in the graph.  From the immunoblot results of Fig. 4 D, it is unclear that there is a reduction in levels of Nrf2 in SOD1-ALS.  
  4. Some of the bands on the immunoblots of Fig. 6 are too intense for quantification by densitometry. For instance many of the bands for TIMM23 are too dense for quantification. Lower exposure would be necessary. There are also great variations in levels of TIMM23 in control samples. If the same amount of protein (25 ug) was loaded in each lane, why the GAPDH bands appear more intense in samples from sALS and SOD1-ALS as compared to controls? This suggests that GAPDH may be upregulated in ALS samples. Thus, normalization to GAPDH may not be the best strategy to quantify the levels of selected proteins.
  5. The manuscript was not well prepared. There are many typing mistakes.

-Cytokine rather than citoquine in the abstract

-Fig. 1 F legend: Quantitative measurement of glycolytic capacity ang glycolytic: should be and.

-Line 199 sALS lymphoblasts showed increased mRNA expression levels of NRF2-parthway. Pathway.

-Line 246 we next studies autophagy in the ALS samples: should be studied.

-Line 259 LC3-interactiv region: should be interactive.

-Line 422 6 sporadic ALS patients (age range 421 55-76, 3 males and 3 males). Should be 3 males and 3 females according to the Table.

-In Fig. 4 A and B. Lamin B used to normalized. If so this should be indicated in the legend.

-Line 341 ‘Interestingly, heme oxygenase-1 (HO-1) is an enzyme that catalyzes the degradation of heme…’ Why interestingly?

Author Response

Response to Reviewer # 2

  • Considering the possibility to work with iPSC-derived motor neurons from ALS patients, lymphoblasts are not the ideal cell type to compare defective molecular mechanisms causative of ALS disease

We agree that iPSC-derived motor neurons would be the ideal cell model, and indeed we are now initiating a collaboration with Dr. Gascon (Cajal Institute, Madrid) to move forward to generate motor neurons from some of the patients involved in our study. Nevertheless, we have been used immortalized lymphocytes from patients suffering from neurodegenerative disorders for more than a decade. LCLs are considered a useful model to study disease mechanisms as they recapitulate the main pathological features of the affected neurons (References 3-5 of the revised version of our Ms). These lymphoblastoid cell lines provide an unlimited supply of material suitable for a number of assays, therefore avoiding re-sampling in vulnerable individuals as ALS patients. On the other hand, we considered these cell lines as a convenient platform for preclinical evaluation of drug candidates for ALS treatment.

  • The number of ALS patients in the sporadic (6) and familial SOD1 (4) is very low. Considering the large variations in some of the results including in the controls, the significance of the findings is questionable

We are aware of the limitation of our work, due to the small number of subjects involved. However, we described, for the first time, distinct alterations in ALS-derived lymphoblasts in carriers of SOD1 mutations compared with control cells, which indicates the need for a correct stratification of patients for clinical care and designing of novel drug trials. Further work with larger population of patients is needed, before these group-specific changes can be considered suitable trait disease markers. Moreover, it remains to be demonstrated the potential of these biomarkers to track the efficacy of personalized treatments in ALS patients. A paragraph highlighting this point has been added to the conclusion section of the revised version of our Ms.

Moreover, each experiment has been performed at least twice, independently, to have consistent evidence of the results obtained.

  1. The data are generally of low quality. In Fig. 4B, the immunoblot suggests great variations in levels of HO1 which is not reflected in the quantification of the graph. I see great variation in the band intensities in controls and sALS. In the controls the variation is not reflected in the graph.  From the immunoblot results of Fig. 4 D, it is unclear that there is a reduction in levels of Nrf2 in SOD1-ALS.  

We have tried to solve the problem of the blots quality. Although there is indeed a variability between the different lymphoblast lines, the experiments have been repeated several times, to be totally sure of the results presented. All samples have been quantified and normalized to housekeeping protein (Lamin B or β-actin). We have included 3 samples from each case, so that the results that we have obtained in the quantifications can be better appreciated in the blots.

  1. Some of the bands on the immunoblots of Fig. 6 are too intense for quantification by densitometry. For instance many of the bands for TIMM23 are too dense for quantification. Lower exposure would be necessary. There are also great variations in levels of TIMM23 in control samples. If the same amount of protein (25 ug) was loaded in each lane, why the GAPDH bands appear more intense in samples from sALS and SOD1-ALS as compared to controls? This suggests that GAPDH may be upregulated in ALS samples. Thus, normalization to GAPDH may not be the best strategy to quantify the levels of selected proteins.

GAPDH is often used as a loading control and, as far as we know, changes in GAPDH protein levels associated to ALS disease have not been reported. In any case, we have repeated the band quantification with a lower exposure of the blots, obtained very similar results. The former Fig. 6 has been modified to accommodate the lower exposure blots and the new densitometric analysis regarding TIMM23 protein.

  1. The manuscript was not well prepared. There are many typing mistakes.

We have tried to correct all the typing mistakes.

Reviewer 3 Report

The paper by Lastres-Becker et al. described the use of lymphoblastoid cell lines obtained from ALS patients (either sporadic or familiar) as a model system to investigate the involvement of well known metabolic and energetic alterations in the pathology. 

Overall the manuscript is well written and the results are well presented, however I would like to see addressed some points, described below, to improve the overall quality of the research.

Title. I think that the title should be modified to better highlight the focus of the manuscript. This paper shows the use of one cell model system (patient-derived lymphoblastoid cell lines) to investigate disease mechanisms. The sentence "cellular models" is too generic. Moreover, the results highlight some differences between the cell lines obtained from sporadic versus one type of familial patients (SOD1-related). However, the number of assessed individuals is too small to support the concept of personalized medicine; the results from this paper only highlight the evidence of some genotype-specific metabolic alterations.

Introduction. The authors should add a paragraph describing the evidence supporting the use of lymphoblastoid cell lines, rather than primary lymphocytes to model disease mechanisms.

Results.

Figure 1. Seahorse Experiments. The authors say that a pool of n=4 lymphoblastoid cell lines per condition (healthy, sALS or fALS patient) were used in the experiment. The advantage of using lymphoblastoid cell lines is that you can culture and expand the cells as needed, so I assume that the experiment could have been performed on n=4 different individual cell lines per condition rather than making a pool. Can the author clarify and justify the reason for the choice of pooling different cell lines?

Methods. Table 1.

The table shows that 13 healthy subjects were used (7 females and 6 males), however in the text (line 423) the authors mention that 14 healthy subjects were used. Please correct.

I think that it would be useful to describe in table one also the stage of disease (e.g. ALS-FRS) and treatment (riluzole? other drug?) of ALS patients at the time when the blood sample was taken.

Moreover, the clinical presentation of some patients is different, i.e. some patients display bulbar pathology whereas others show spinal pathology and one show respiratory pathology. The definition of "respiratory" is not clear, the authors should describe better the clinical classification of this patient. Moreover, all the SOD1 carrier display spinal pathology. Since only some of the 6 sALS patients lymphoblastoid cell lines were used  in the experiments (n=4 Fig. 1A, n=5 Fig. 1B, n=5-6 in Fig. 3 experiments, n=5 in Fig. 4, only n=4 in Fig. 6 where high variability in the autophagic flux was reported), I think that the authors should verify whether the sporadic patients used in each and every experiment shown are homogeneous in term of disease presentation (bulbar, versus spinal or respiratory). In fact, one could argue that some differences between sALS and fALS cell lines could be linked to the type of clinical presentation (bulbar versus spinal) rather than the genetics of the disease (sporadic versus SOD1-related) given that the clinical presentation is not homogeneously distributed among sporadic and familial cases. Please address this comment and clarify.

Finally, at lines 431-433 the authors say that the "individuals positive for mutations in TARDBP, FUS and C9orf72 were excluded for this work". It is not clear whether the sporadic ALS were only tested for these 4 genes or whether they were also tested for any of the many other well known genes associated with familial ALS (such as ALS2, VAPB, CHMP2B, SETX ecc..).

Conclusion.

Please expand this paragraph to better highlight critical insights from this study that could instruct further research oriented to personalized medicine approaches. For instance...could the patient derived lymphoblastoid cell lines be used to verify the response to treatment of a patient rather than just highlight differences between sporadic and familial ALS cases in some disease mechanisms?

Minor comments.

Some spell and typos checking throughout the text is recommended.

Some examples are provided below:

Abstract, line 30: Please correct "citoquines" with "cytokines".

Introduction, line 58: "focused in" should be changed in "focused on"

Line 246: "studies" should be corrected with "studied"

Author Response

Responses to Reviewer #3

Title. I think that the title should be modified to better highlight the focus of the manuscript. This paper shows the use of one cell model system (patient-derived lymphoblastoid cell lines) to investigate disease mechanisms. The sentence "cellular models" is too generic. Moreover, the results highlight some differences between the cell lines obtained from sporadic versus one type of familial patients (SOD1-related). However, the number of assessed individuals is too small to support the concept of personalized medicine; the results from this paper only highlight the evidence of some genotype-specific metabolic alterations.

Following the Reviewer advise we have changed the title to “Molecular Alterations in Sporadic and SOD1-ALS immortalized lymphocytes: Towards a Personalized Therapy”

Introduction

The authors should add a paragraph describing the evidence supporting the use of lymphoblastoid cell lines, rather than primary lymphocytes to model disease mechanisms

Following the reviewer suggestions, a paragraph has been added to the Introduction section (lines 50-54) to justify the use of lymphoblastoid cell lines rather than fresh lymphocytes.

Seahorse experiments

Initially, we performed a set de experiments with pooled cell lines in the Seahorse system and RMN. Later on, we checked the Seahorse findings by doing a new set of experiments with individual cell lines obtaining similar results, as it can be appreciated in the  figure below.

Parameters of respiratory in control (n=4) and patient-derived (sporadic or SOD1-mutant (n=4) lymphoblasts. Bars depict grouped control and patients’ values. Values shown are the mean ± SEM

In accordance with the pooled samples, a trend to have increased basal respiration is observed in ALS samples, being the most significant change the reduced spare respiratory capacity of both sporadic and SOD1-mutant ALS lymphoblasts.

We decided to keep the initial set of Seahorse experiments to be able to correlate the ECAR rates with the levels of lactate in the extracellular medium, assessed by H1RMN in the pooled samples.

The table shows that 13 healthy subjects were used (7 females and 6 males), however in the text (line 423) the authors mention that 14 healthy subjects were used. Please correct

We appreciated this observation. A control subject was omitted in the Table. It has been now added.

I think that it would be useful to describe in table one also the stage of disease (e.g. ALS-FRS) and treatment (riluzole? other drug?) of ALS patients at the time when the blood sample was taken.

We thank the Reviewer for this advice. We have now included in the Methods section (lines 438-445) pertinent information about the genetic analysis performed, the ALSRF-r values and treatments when available,

Moreover, the clinical presentation of some patients is different, i.e. some patients display bulbar pathology whereas others show spinal pathology and one show respiratory pathology. The definition of "respiratory" is not clear, the authors should describe better the clinical classification of this patient. Moreover, all the SOD1 carrier display spinal pathology.

Respiratory onset is a presenting symptom in a reduced number of ALS patients (Shoesmith et al., 2007. J. Neurol. Neurosurgery Psichiatry 78, 629). In these patients the limb muscle motor neurons are less sever involved.

Most of the cell lines from sporadic cases had been used in our laboratory to study TDP-34 homeostasis among another processes (Posa et al. Molecular neurobiology 2019, 56, 2424-2432; Vaca et al., Journal of neurochemistry, 10.1111/jnc.15118, 2020). We never found significant differences related to different clinical presentation. Up to now we do not have SOD1-mutants with bulbar onset, however it is our intention to increase our cell line collection with more familial cases.

Finally, at lines 431-433 the authors say that the "individuals positive for mutations in TARDBP, FUS and C9orf72 were excluded for this work". It is not clear whether the sporadic ALS were only tested for these 4 genes or whether they were also tested for any of the many other wellknown genes associated with familial ALS (such as ALS2, VAPB, CHMP2B, SETX ecc..).

We appreciated this remark. The pertinent information is now provided in lines 438-441 of the revised version of our Ms.  In brief, genetic testing for SOD1, TARDBP, FUS and C9orf72 was performed in all the sporadic cases. Two SOD1-mutant cases were analyzed with a panel including 48 ALS-related genes, while the full exome of the remained cases were sequenced.

Conclusion.

Please expand this paragraph to better highlight critical insights from this study that could instruct further research oriented to personalized medicine approaches. For instance.. could the patient derived lymphoblastoid cell lines be used to verify the response to treatment of a patient rather than just highlight differences between sporadic and familial ALS cases in some disease mechanisms?

 We had added a final paragraph to the discussion section to summarize our findings and to highlight their relevance for stratification of patient for clinical care and design of drug trials.  In addition, the Conclusions section has been re-written

Minor comments

Typos errors have been corrected

Round 2

Reviewer 1 Report

I thank the authors for their reply to my concerns, but I do not feel to suggest the publication of this work.

  1. The authors say: "The ultimate goal of our work is to open avenues for disease-modifying therapies in ALS patients in a personalized manner based on potential differences in molecular disease mechanisms that might exist between sporadic or familial cases." but in their work they just showed that some pathways resulted altered in LCLs, without providing that LCLs would be an optimal cell model to test new treatments.
  2. In the results section the authors did not explain that with their experiment, they showed only the increase in the autophagic carrier flux (to the stage of cargo reaching the lysosome). They could not state that the whole autophagic flux is increased (that in case of ALS is inhibited at the degradation step). 
  3. Regarding the Seahorse experiment, authorr are right. In Pansarasa paper it is not clear the number of cell seeded. However, the ECAR = 500 pmol/min at basal level is extremely high, thus I am not sure that the experiment was carried out correctly. Regarding cytokines experiment, the authors reported article of 1997 and 2009. In the last years, more sensitive and reliable methods were used to evaluate cytokines level.
  4. As stated by the authors "we have focus on investigating some of the well-known molecular pathways involved in the ALS etiology", and this is why I think that the article lacks of originality and novelty. I also disagree with this sentence "our results can help to stratification of patients for distinct therapeutic strategies in each case", because the stratification between sALS and ALS-SOD1 is obviously genetics.

Reviewer 2 Report

The paper has merits and the authors have adequately address reviewer's points.